# Investigation on Fatigue Performance of Asphalt Mixture Reinforced by Basalt Fiber

**DOI:** 10.3390/ma14195596

**Published:** 2021-09-26

**Authors:** Keke Lou, Xing Wu, Peng Xiao, Cong Zhang

**Affiliations:** 1College of Civil Science and Engineering, Yangzhou University, Yangzhou 225100, China; lkkyzu@163.com (K.L.); mx120190452@yzu.edu.cn (X.W.); 006482@yzu.edu.cn (C.Z.); 2Research Center for Basalt Fiber Composite Construction Materials, Yangzhou 225127, China

**Keywords:** asphalt mixture, basalt fiber, fatigue performance, cumulative dissipated energy, change rate of dissipated energy

## Abstract

Basalt fiber has been widely used in asphalt mixture due to its excellent mechanical properties and good combination with asphalt. In order to systematically evaluate the enhancement effect of basalt fiber on the fatigue performance of the mixtures, gradations of Stone Mastic Asphalt and Superpave with different nominal maximum aggregate sizes, namely SMA-13, SUP-20 and SUP-25, were prepared, and a four-point bending beam fatigue test was adopted under the strain control mode. The fatigue damage mode was assessed based on the phenomenology theory, energy dissipation theory and change rate of dissipated energy. The results showed that basalt fiber could well increase the fatigue life of the mixtures. Basalt fiber could also increase the cumulative dissipated energy of the mixtures, and it was linearly correlated with the fatigue life in double logarithmic coordinates. In the meantime, adding basalt fiber could increase the change rate of dissipated energy of the mixtures. Furthermore, it is not appropriate to take the stiffness modulus declined to 50% of the original as the fatigue failure criterion of the mixture; this paper suggested that it is reasonable when the stiffness modulus was 15–25% that of the initial. These findings provide a theoretical basis for exploring the fatigue failure of asphalt pavements.

## 1. Introduction

Presently, owing to increases in traffic volume and the number of heavy vehicles, pavement materials are required to have more strength and durability during the service time because there are more threats to the pavement [1,2]. Under this condition, the fatigue performance or the deformation of the asphalt mixtures caused by fatigue damage are the important factors needed to be considered in the engineering practice [3,4,5]. Several measures were adopted to enhance the performance of the pavement [6,7,8,9,10]. The previous studies showed that adding fibers into the mixtures is an effective way [11,12]. Generally, the fibers added in the asphalt pavement could be divided into several categories such as the plant fibers (lignin fiber and bamboo fiber, etc.), the synthetic polymer fibers (polyester fiber, etc.) and the mineral fibers (glass fiber and basalt fiber, etc.) [13,14,15].

Different fibers have different characteristics, for instance, plant fibers usually have less strength than mineral fibers. The fracture strength of lignin fibers (plant fiber) is smaller than basalt fibers (mineral fiber) [16], it could also degrade in the mixtures during long-term service. However, the plant fibers such as the lignin fiber could absorb the asphalt very well [17,18] in the mixtures. Synthetic polymer fibers usually consume a lot of energy in the manufacturing process and the by-products usually have some negative effects on the environment. Mineral fiber is a kind of fiber of which the raw material is the rock. Basalt fiber (BF) is a kind of mineral fiber, and it has excellent mechanical properties such as high tensile strength and high elastic modulus, etc. [19]. The chemical or physical stability of basalt fiber is also very good [20,21,22,23]. Therefore, it could act as a very stable additive in the strengthening of the pavement materials. The production process of basalt fiber is also very eco-friendly [24].

The fatigue performance of the asphalt mixtures could be evaluated using several methods and could be divided into several categories. The first one is the fatigue test conducted based on the realistic road load conditions. However, it could easily be affected by the temperatures; in the meantime, the test parameters such as the moisture, the temperature, etc. could not be precisely controlled if we need to conduct the test according to the real temperature or other kinds of data collected from the real environment of a specific road. The second kind of fatigue test is the full-scale loop test [25], which needs a lot of energy or money. Since the full-scale loop test is carried out on the real road and the load is relatively similar to the real vehicle loads, it is more accurate. 

There are also some fatigue tests that the researchers could easily conduct in the laboratories. These tests are listed as follows: the four-point bending fatigue test (4PB) [26,27], the two-point cantilever fatigue test (2PB) [28], the direct tensile fatigue test (UTFT) [29,30] and the indirect tensile fatigue test (IDT) [31,32], etc. The Strategic Highway Research Program issued the “Fatigue Response of Asphalt-Aggregate Mixes (SHRP-A-404)” [33] after comparing different test methods of the fatigue tests of the asphalt mixtures. The report suggested the four-point bending fatigue test as the standard test method to evaluate the fatigue performance of asphalt mixtures. The results of this test are very reliable and could be repeated easily. There are some researchers conducting studies about the fatigue performance of asphalt mixtures using the four-point bending fatigue test. Zhu et al. [34] used this method to test the fatigue life of the aged and unaged asphalt mixtures and found that the fatigue test curve of the aged mixture is very similar to that of the unaged mixture. In the meantime, the fatigue life is far lower than that of the unaged sample. Huang et al. [35] adopted this test method to assess the fatigue performance of the modified asphalt mixture and found that the content of the modifier could well affect the fatigue performance of the fatigue life of the mixtures. Han et al. [36] utilized different test indexes in the four-point trabecular bending fatigue test to evaluate the fatigue performance of the asphalt mixtures and concluded that the different test indexes of this test are in good correlation with the fatigue life, which is decided when the stiffness modulus was 50% that of the initial stiffness modulus. However, there is still a lack of adequate research in the mixtures modified by basalt fiber, and the evaluation index and theory are relatively deficient.

The primary objective of this work was to evaluate the enhancement effect of basalt fiber on the fatigue performance of the mixtures. Therefore, the commonly used gradations of the different pavement layers were adopted to test the fatigue performance of these mixtures with and without basalt fibers (SMA-13, SUP-20 and SUP-25). Different analysis methods of the four-point bending fatigue test were adopted, including the phenomenology theory, energy dissipation theory and change rate of dissipated energy. The modified fatigue life decision criterion of the fatigue life was also given in this study to decide the fatigue life more accurately. This paper has certain significance in guiding the evaluation and the strengthening effect of the fatigue performance of the basalt fiber-reinforced asphalt mixtures.

## 2. Materials and Methods

### 2.1. Materials

#### 2.1.1. Basalt Fiber and Lignin Fiber

The chopped basalt fiber used in this paper was bought from Jiangsu Tianlong Basalt Fiber Co., Ltd., Yizheng, China. The lignin fiber adopted was provided by JRS company of Germany, Rosenberg, Germany. The macro and the micro images are shown in Figure 1 and Figure 2. The technical performances of the chopped basalt fiber and lignin fiber are listed in Table 1. 

#### 2.1.2. Asphalt

The pure asphalt used in this paper was bought from the Sinopec Group, China. SBS modified asphalt was provided by Jiangsu Tiannuo Road Materials Technology Co., Ltd., Zhenjiang, China. The properties of the pure asphalt and the SBS modified asphalt are summarized in Table 2 and Table 3, which are in accordance with the standard test methods of bitumen and bituminous mixture for highway engineering, JTG E20-2011 [37].

#### 2.1.3. Mixture Design and Experiment Scheme

The gradation curves of SMA-13, SUP-13 and SUP-25 asphalt mixtures are shown in Figure 3. The gradation curves were conducted according to Technical Specification for Construction of Highway Asphalt Pavements (JTG F40-2004) [38]. Basalt aggregates and limestone aggregates were selected for SMA and SUP, the density of the aggregates is 2.927 and 2.698 g/m^3^, respectively. The details of the asphalt mixture samples were shown in Table 4, the properties of the mixture meet the requirements of JTG E20-2011 [37].

### 2.2. Test Method and Fatigue Test Index

#### 2.2.1. Four-Point Bending Fatigue Test

Four-point bending fatigue test was utilized to evaluate the fatigue performance of the asphalt mixtures. The length of the sample was 380 mm ± 5 mm, the width of the sample was 63.5 mm ± 5 mm and the height of the sample was 50 mm ± 5 mm. As shown in Figure 4, the samples were cured at 15 °C according to standard test methods of bitumen and bituminous mixtures for highway engineering (JTG E20-2011) [37]. The test was preloaded for 50 cycles under the target strain level and then the initial stiffness modulus was tested and collected. When the initial stiffness modulus declined to 50% of the original value, the test was ended. The loading frequency was 10 Hz. In order to comprehensively reflect the fatigue performance of the asphalt mixtures, three kinds of strain levels were adopted to test the fatigue life of the samples under the strain control mode. Therefore, the strain was adopted at 450, 650 and 850 με in the test. In this study, three typical specimens were selected for each gradation in order to obtain enough effective statistical data.

#### 2.2.2. Test Index Based on Phenomenology Theory

The fatigue index based on the phenomenology theory used in this paper is the fatigue life, because it could directly reflect the fatigue performance of the test samples. In the strain control mode, the specimen was regarded as damaged when the initial stiffness modulus become 50% or lower that of the initial value. The results of the test were represented in Equation (1). In this equation, N_f_ is the loading number during the whole fatigue test, and c and m are the constants that depend on the composition and properties of the asphalt mixture. ε is the maximum strain applied to the mixtures.
(1)$$N_{f}=c{\left(\frac{1}{\varepsilon }\right)}^{m}$$

#### 2.2.3. Test Indexes Based on Energy Dissipation Theory

The test indexes based on energy dissipation theory are the dissipative angle and the cumulative dissipated energy. The peak of the strain lags behind the stress peak because an asphalt mixture is a kind of viscoelastic material. The dissipative angle is the time difference when the stress and strain become the biggest. The calculation equation of the dissipative angle is shown in Equation (2), Where φ(N) is the dissipative angle when the loading number is *N*. ω is the loading frequency. △t(N) is the time of the strain peak lagging behind the peak stress when the loading number is *N*. The cumulative dissipated energy, $W_{Nf}$, is calculated using Equation (3). $△W_{N}$ is the dissipated energy when the number of the loading is N. The calculation of $△W_{N}$ is shown in Equation (4).
(2)φ(N)=ω△t(N)=2πf△t(N)
(3)$$W_{Nf}=\sum_{i=1}^{Nf}△W_{i}$$
(4)$$△W_{N}=\pi \sigma_{t}\left(N\right)\varepsilon_{t}\left(N\right)\sin\left[\phi \left(N\right)\right]$$

#### 2.2.4. Test Index Based on the Change Rate of Dissipated Energy

The change rate of dissipated energy (DERC) is the ratio of dissipated energy generated in two adjacent loading cycles to that generated in the previous cycle. It is calculated by Equation (5). According to the relationship between the change rate of dissipated energy and the loading cycles, the curve of the change rate of dissipated energy could be divided into three stages, namely, rapid descent stage, stable stage and rapid growth stage. The stable change rate of dissipated energy in the second stage is called the stable value of the DERC and is marked as PV. $DE_{a}$ and $DE_{b}$ are the dissipated energy when the loading cycles are a and b, respectively.
(5)$$DERC=\frac{DE_{a}-DE_{b}}{DE_{a}\times \left(b-a\right)}$$

## 3. Results and Discussion

### 3.1. Fatigue Analysis Based on Phenomenology Theory

The fatigue life of the mixture is illustrated in Figure 5. It is clear that compared with the control group, the fatigue life N_f,50_ of SMA-13 with basalt fiber increased by 51, 185 and 166% at the strain level of 450, 650 and 850 με, respectively. Similarly, the N_f,50_ of SUP-20 with basalt fiber increased by 230, 73 and 4% and the N_f,50_ of SUP-25 with basalt fiber increased by 362, 95 and 45%. It visibly indicates that basalt fiber can significantly enhance the fatigue resistance of the asphalt mixture. The explanation could be that the high elastic modulus and superior fracture elongation of basalt fiber can improve the recovery ability of asphalt deformation and effectively prevent the propagation of fatigue fracture.

The log N_f,50_ index obtained from different strain levels is plotted in Figure 5d and all the regression coefficients are gathered in Table 5, and the equations pass the statistical significance check (α=0.05). It can be seen that the slopes of the fatigue lines went up by 11.7, 36.4 and 94.3% for SMA-13, SUP-20 and SUP-25, respectively, which indicates that fibers could enhance the fatigue life of the asphalt mixture more effectively at relatively low strain levels. Moreover, the intercepts of the fatigue lines increased by 9.5, 14.7 and 36.6%, implying that fibers could improve the fatigue resistance of the asphalt mixture by decelerating the fatigue decay rate, which is consistent with reference [11].

In addition, compared with the samples of C0, C1 and C2, the application of SBS modified asphalt can also improve the fatigue performance of the asphalt mixture. Furthermore, the addition of basalt fiber can obtain better reinforcement effect than SBS modified asphalt.

### 3.2. Fatigue Analysis Based on Energy Dissipation Theory

#### 3.2.1. Dissipative Angle

The dissipative angle is represented by the average value of all the dissipative angles during the fatigue test. The dissipative angles of all the test samples are shown in Figure 6. It could be concluded that the dissipative angle increased gradually when the strain level increased. This is because when the strain level increases, there will be less stress needed to reach the same deformation. Therefore, the viscoelastic property of the mixtures will be higher. The influence of basalt fiber on the dissipative angle is not very obvious.

#### 3.2.2. Dissipated Energy

The dissipated energy (DE) of the samples with or without basalt fiber under different train levels is shown in Figure 7. It could be clearly seen from the figure that the dissipated energy decreased gradually when the loading cycles increased. Additionally, the dissipated energy decreased very fast in the beginning and then it becomes relatively stable after the rapid decrease. The results showed that with the increase in loading cycles, the energy required to reach the same strain is smaller. After adding basalt fibers into the mixtures, the stable part in the testing curve was maintained for a longer time, which means that basalt fiber could well increase the fatigue life and the toughness of the mixtures. In the meantime, the dissipated energy of all the samples increased after adding basalt fiber. This is mainly because that basalt fiber could bear some of the stress in the mixtures, making the dissipated energy that was needed to reach the same strain bigger.

#### 3.2.3. Cumulative Dissipated Energy

Figure 8 shows that basalt fiber could increase the cumulative dissipated energy (CDE) of all the asphalt mixtures. The CDE value of SMA-13 mixtures under 450, 650 and 850 με increased by 53, 180 and 155% after adding basalt fiber. The relative increasing percentages of SUP-20 are 76, 97 and 89% and the increasing extents of SUP-25 are 420, 77 and 65%. Thus, the effects of basalt fiber on the different kinds of asphalt mixtures are different. When the nominal maximum aggregate size (NMAS) is 13 mm (SUP-13), which is the relatively lower size, the effect of basalt fiber on the CDE value is the biggest when the strain level is 650 με, followed by 850 and 450 με. When the NMAS is 20 mm (SUP-20), the effect of the basalt fiber on the CDE is similar to that of the SMA-13. However, the influence effect of basalt fiber under these three strain levels is very similar. When the NMAS is 25 mm (SUP-25), the enhancement effect of basalt fiber on it is the best when the strain level is 450 με, followed by 650 and 850 με. Generally, when the NMAS is 13 mm, basalt fiber could better enhance the fatigue performance under bigger strain levels. When the NMAS is 25 mm, basalt fiber could better enhance the fatigue performance under smaller strain levels. When the NMAS is 20 mm, the influence of basalt fiber on the fatigue property of the mixtures is similar under different strain levels.

The fitting results between lgW_Nf_ and lgN_f_ are shown in Figure 9. It is clearly shown that there is a good correlation between lgW_Nf_ and lgN_f_; the R-squared exceeds 0.946. This means that there is a good linear relationship between the cumulative dissipated energy and the fatigue life in double logarithmic coordinates. Furthermore, according to the statistical test results (α=0.05), F = 126.851 > 4.381 = F _0.05_ (19), which indicates that the equation passes the statistical significance check.

### 3.3. Fatigue Analysis Based on Change Rate of Dissipated Energy

The stable value PV was calculated according to the four-point bending fatigue test results of the asphalt mixture under strain control mode, which is shown in Figure 10. It is obvious that basalt fiber could increase the PV value of the mixtures under different strain levels. It means that the energy loss of the asphalt mixture could well increase after adding basalt fiber. Moreover, the stable PV value of the DERC increases gradually with the increase in the strain level. This is due to the fact that the energy gradient during different loading cycles is larger when the strain level is larger, which causes greater damage to the asphalt mixture.

Furthermore, taking SUP-20 as an example, the change rate of dissipated energy (DERC) of the samples is listed in Figure 11. As is mentioned before, the DERC should have three stages. However, Figure 11 shows that the experiment was stopped when the stiffness modulus became 50% of the initial stiffness value, the DERC was still in the second stage, which is the stable stage. It means that no fatigue damage occurred to the specimen. At this time, the fatigue life does not represent the real fatigue characteristics of the material. Therefore, further analysis to explore more suitable criteria to determine the ending time of the fatigue test is necessary.

### 3.4. Further Analysis about the DERC

Taking SMA-13 and SUP-20 with and without basalt fiber as examples, the four-point bending fatigue test was carried out under the strain of 850 με. The test terminated when the stiffness modulus decreased to 10% of the initial value. The results of the DERC are listed in Figure 12. The results showed that the stiffness modulus of the critical points in the second and third stages is much later than 50% of the initial value, which further proves that it is not appropriate to take the stiffness modulus decreased to 50% of the initial value as the termination condition of the test. Furthermore, the stiffness modulus of the critical point is listed in Table 6. It is clear that the stiffness modulus of the critical point is less than 50% of the initial value, and the corresponding fatigue life is larger. Specifically, it is more reasonable to decide the ending time of the fatigue test when the stiffness modulus was 15–25% that of the initial value.

## 4. Conclusions

The primary objective of this work was to evaluate the enhancement effect of basalt fiber on the fatigue performance of the mixtures; gradations of SMA-13, SUP-20 and SUP-25 were prepared, four-point bending beam fatigue test was adopted and the fatigue damage mode was assessed based on the phenomenology theory, energy dissipation theory and change rate of dissipated energy. The following conclusions can be derived, which could provide a theoretical basis for exploring the fatigue failure of asphalt pavements.

Basalt fiber can significantly enhance the fatigue life of asphalt mixture. The addition of basalt fiber can obtain a better reinforcement effect than SBS-modified asphalt. The fatigue life has a good linear relationship with the strain levels in the semi logarithmic coordinate.The dissipative angles increase when the strain level increases. The influence of basalt fiber on the dissipative angle is not very obvious.The dissipated energy decreased when the loading cycles increased. After adding basalt fibers, the stable part in the testing curve was maintained for a longer time, and the dissipated energy of all the samples increased. There is a good linear relationship between cumulative dissipated energy and fatigue life in double logarithmic coordinates.Basalt fiber could increase the PV value of the mixtures under different strain levels. When the stiffness modulus became 50% of the initial stiffness value, the DERC was still in the second stage. It is more reasonable to decide the ending time of the fatigue test when the stiffness modulus is about 15–25% that of the initial value.The addition of basalt fiber may increase the cost of pavement construction, and it is more suitable for application in heavy traffic sections.

## Figures and Tables

**Figure 1 materials-14-05596-f001:**
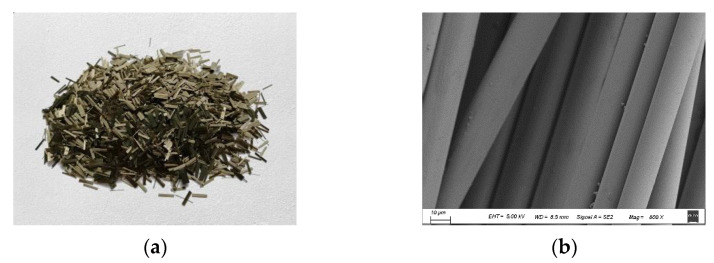
Basalt fiber: (**a**) Macro image; (**b**) Micro image.

**Figure 2 materials-14-05596-f002:**
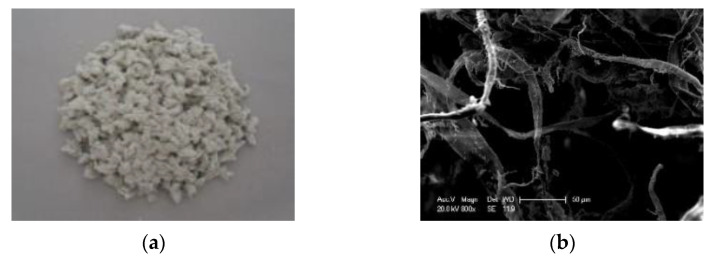
Lignin fiber: (**a**) Macro image; (**b**) Micro image.

**Figure 3 materials-14-05596-f003:**
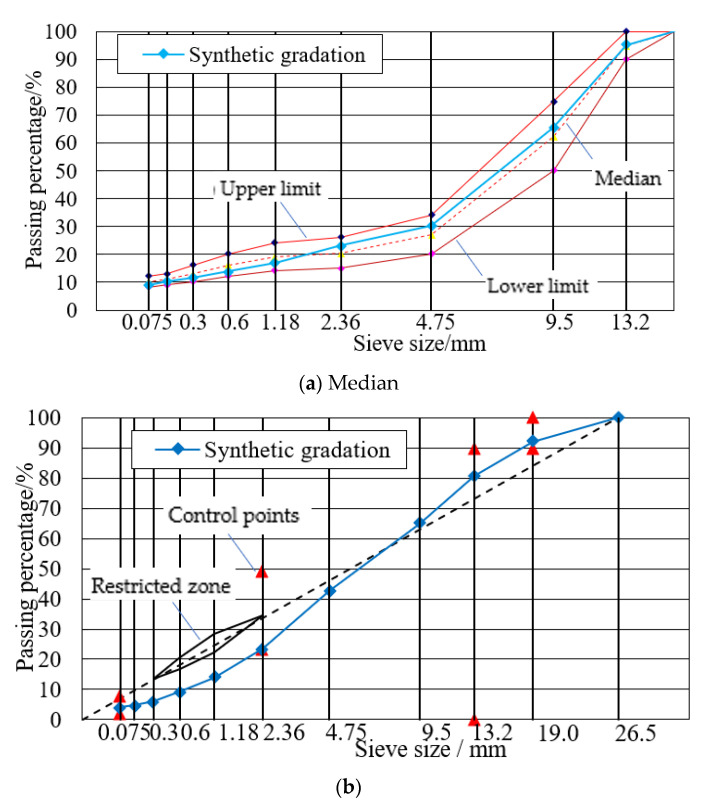
Gradation curves: (**a**) SMA-13; (**b**) SUP-20; (**c**) SUP-25.

**Figure 4 materials-14-05596-f004:**
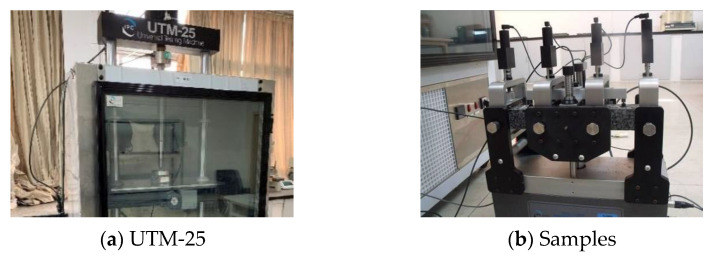
Four-point bending fatigue test.

**Figure 5 materials-14-05596-f005:**
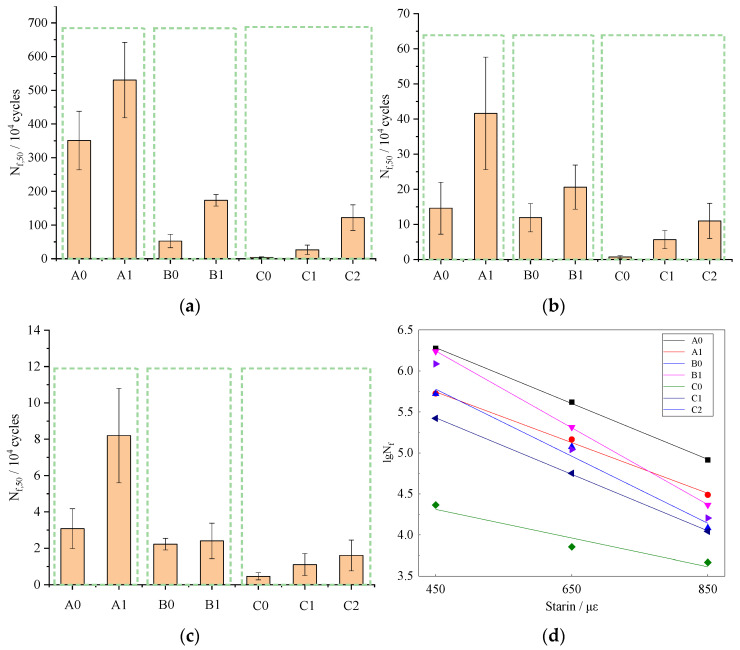
Fatigue test results: (**a**) 450 με, (**b**) 650 με, (**c**) 850 με and (**d**) relationship between fatigue life and strain.

**Figure 6 materials-14-05596-f006:**
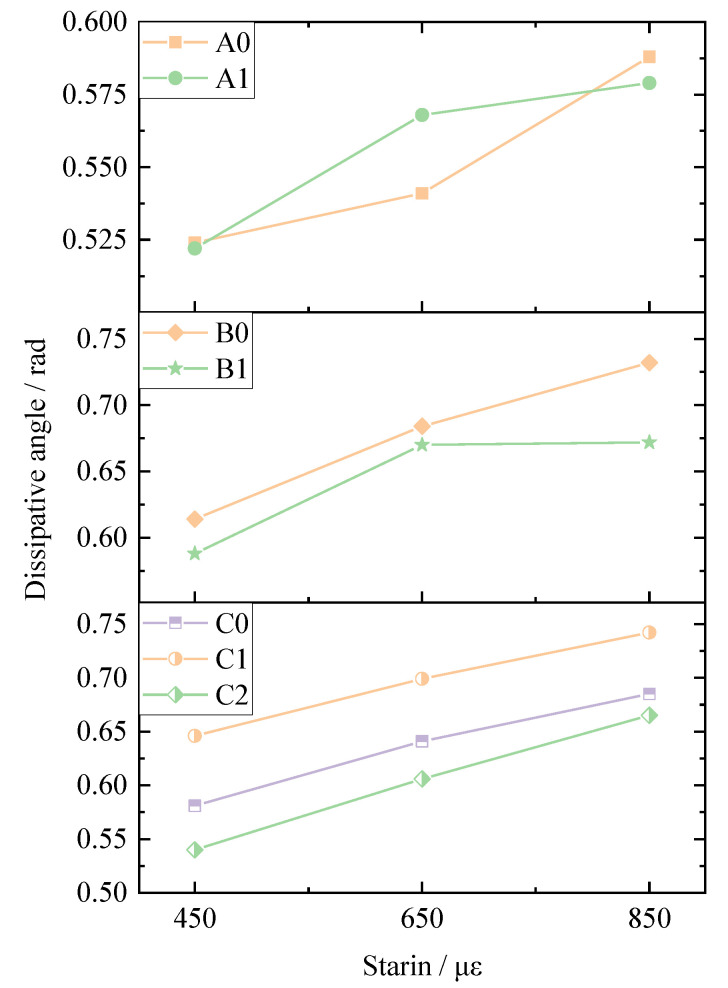
Dissipative angles under different strains.

**Figure 7 materials-14-05596-f007:**
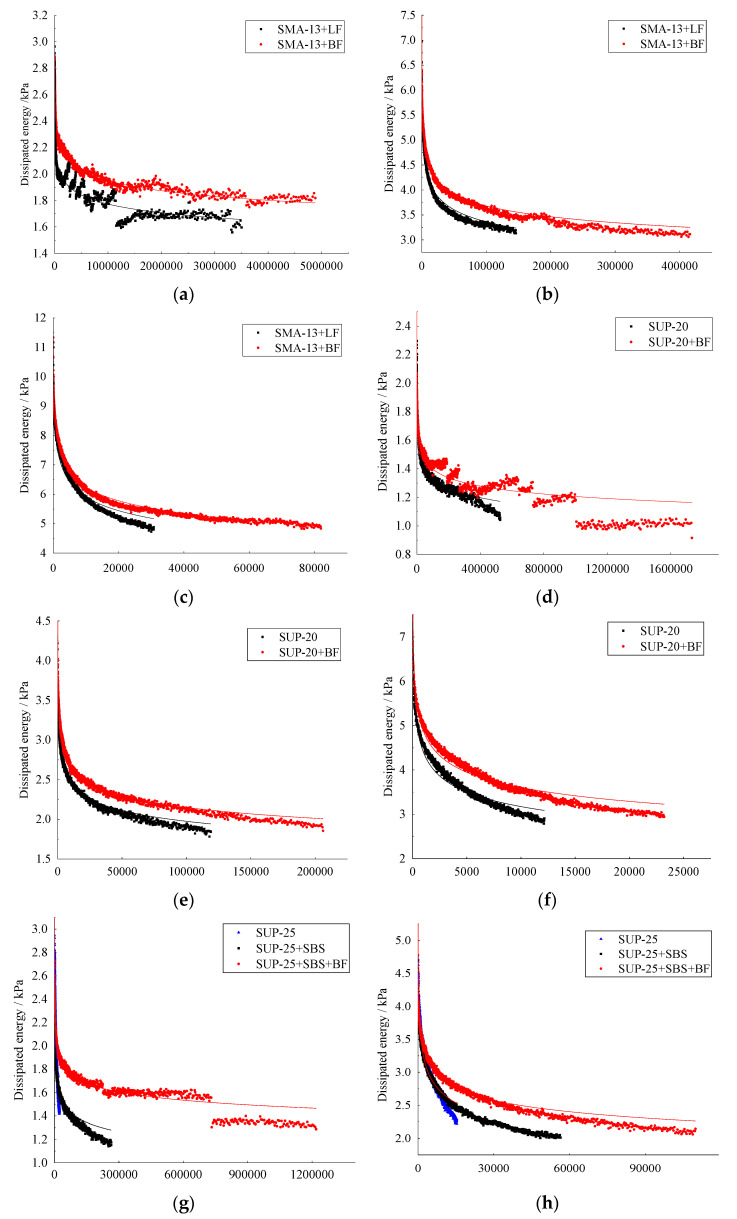
Dissipated energy: (**a**) SMA-13 (450 με), (**b**) SMA-13 (650 με), (**c**) SMA-13 (850 με), (**d**) SUP-20 (450 με), (**e**) SUP-20 (650 με), (**f**) SUP-20 (850 με), (**g**) SUP-25 (450 με), (**h**) SUP-25 (650 με) and (**i**) SUP-25 (850 με).

**Figure 8 materials-14-05596-f008:**
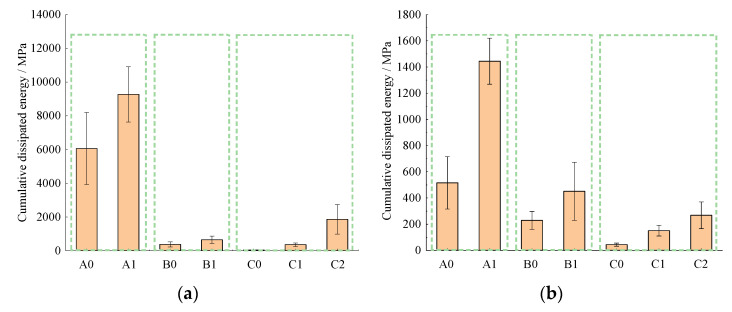
Cumulative dissipated energy: (**a**) 450 με, (**b**) 650 με and (**c**) 850 με.

**Figure 9 materials-14-05596-f009:**
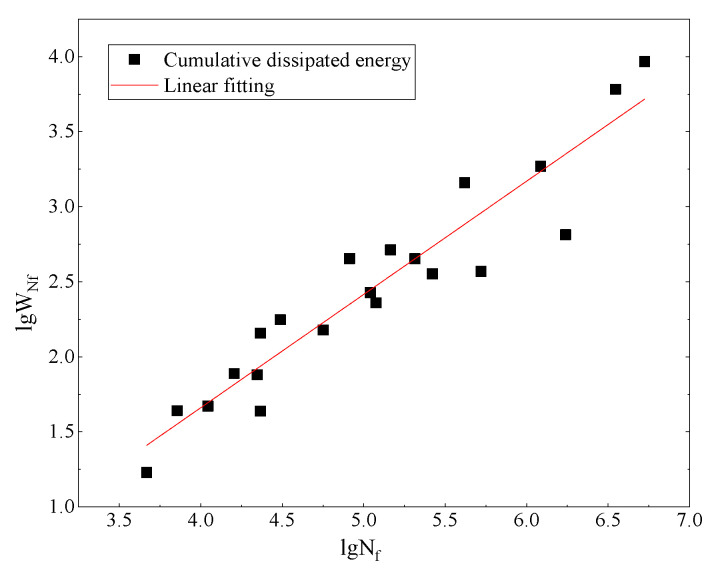
Fitting results.

**Figure 10 materials-14-05596-f010:**
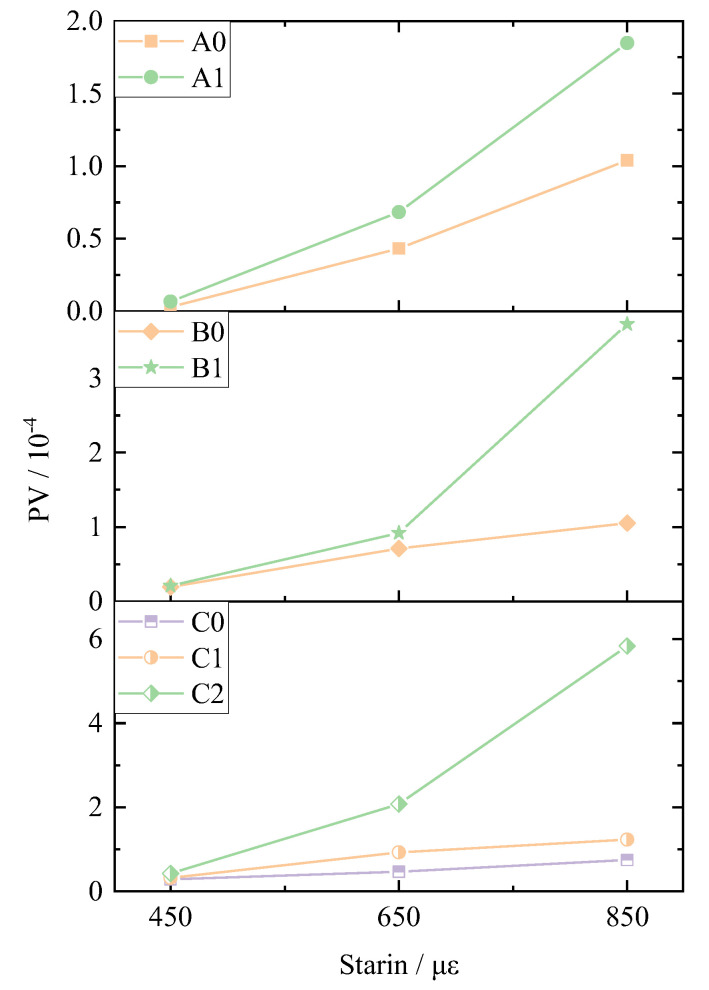
PV of the mixtures.

**Figure 11 materials-14-05596-f011:**
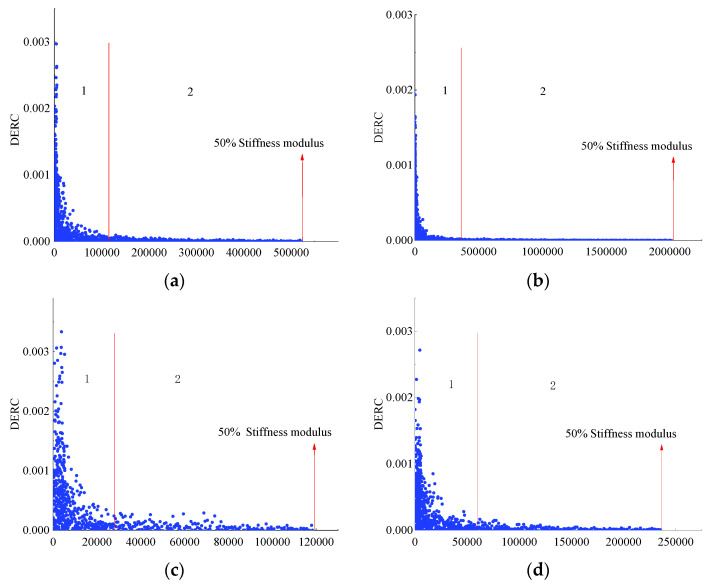
Change rate of dissipated energy: (**a**) SUP-20 (450 με), (**b**) SUP-20 with BF (450 με), (**c**) SUP-20 (650 με), (**d**) SUP-20 with BF (650 με), (**e**) SUP-20 (850 με) and (**f**) SUP-20 with BF (850 με).

**Figure 12 materials-14-05596-f012:**
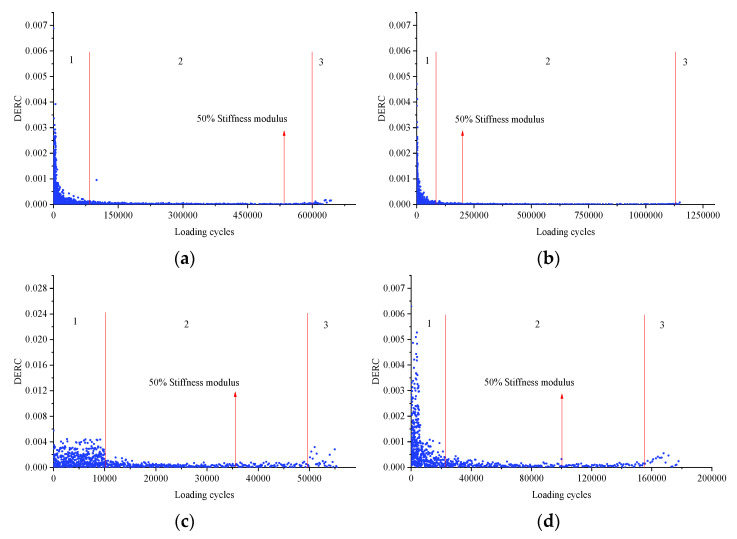
Change rate of dissipated energy: (**a**) SMA-13, (**b**) SMA-13 with BF, (**c**) SUP-20 and (**d**) SUP-20 with BF.

**Table 1 materials-14-05596-t001:** Properties of short-cut basalt fiber.

Index	Basalt Fiber	Lignin Fiber
Density, g/cm^3^	2.710	0.910
Heat resistance, °C	1550	260
Fracture strength, MPa	≥2000	<300
Modulus of elasticity, GPa	100	30

**Table 2 materials-14-05596-t002:** Properties of pure asphalt.

Test Index		Requirement	Result	Test Method
Penetration (25 °C, 100 g, 5 s)/0.1 mm		60~80	71.2	T0604
Penetration Index (PI)		−1.5~1.0	−0.8	T0604
Softening point/°C		≮45	47.1	T0606
Wax content/%		≯2.2	1.8	T0615
flash point/°C		≮260	>300	T0615
Residue after RTFOT	Weight change/%	≯±0.8	−0.02	T0610
	Penetration ratio/%	≮61	61.2	T0604
	Residual ductility (10 °C)/cm	≮6	6.2	T0605
	Residual ductility (15 °C)/cm	≮15	28.4	T0605

**Table 3 materials-14-05596-t003:** Properties of SBS modified asphalt.

Test Index		Requirement	Result	Test Method
Penetration (25 °C)/0.1 mm		60~80	67	T0604
Softening point/°C		≮55	58	T0606
Ductility (5cm/min, 5 °C)/cm		≮30	43	T0605
Penetration Index (PI)		−0.4~1.0	0.3	T0604
Kinematic viscosity (235 °C)/Pa·s		≯3	1.8	T0625
Elastic recovery rate (25 °C)/%		≮65	78	T0662
Softening point difference/°C		≯2.5	1.4	T0661
Residue after RTFOT	Weight change/%	≯±1.0	−0.06	T0610
	Penetration ratio/%	≮60	82	T0604
	Residual ductility (15 °C)/cm	≮20	36	T0605

**Table 4 materials-14-05596-t004:** Details of the asphalt mixture samples.

No.	Gradations	Asphalt	Additive	Content/%	OAC/%	VMA	VFA/%	VV/%
A0	SMA-13	SBS modified asphalt	Lignin fiber	0.3	6.1	17.06	76.03	4.09
A1	SMA-13	SBS modified asphalt	Basalt fiber (6 mm)	0.3	6.0	16.83	75.94	4.05
B0	SUP-20	SBS modified asphalt	/	/	4.3	13.41	70.09	4.01
B1	SUP-20	SBS modified asphalt	Basalt fiber (9 mm)	0.3	4.5	13.47	69.86	4.06
C0	SUP-25	Pure asphalt	/	/	4.2	12.25	66.85	4.06
C1	SUP-25	SBS modified asphalt	/	/	4.2	12.22	66.94	4.04
C2	SUP-25	SBS modified asphalt	Basalt fiber (12 mm)	0.4	4.4	12.36	67.23	4.05

**Table 5 materials-14-05596-t005:** Semilog fatigue equation of different asphalt mixtures.

Types of Asphalt Mixture	Semilog Fatigue Equation	R^2^
A0	$\log N_{f}=7.134-0.00309\varepsilon$	0.9810
A1	$\log N_{f}=7.815-0.00345\varepsilon$	0.9920
B0	$\log N_{f}=7.278-0.00343\varepsilon$	0.9993
B1	$\log N_{f}=8.350-0.00468\varepsilon$	0.9999
C0	$\log N_{f}=5.096-0.00175\varepsilon$	0.9661
C1	$\log N_{f}=5.978-0.00245\varepsilon$	0.9998
C2	$\log N_{f}=8.166-0.00476\varepsilon$	0.9979

**Table 6 materials-14-05596-t006:** Stiffness modulus of the critical point.

Gradations	Stiffness Modulus/MPa	50% Stiffness Modulus/MPa	N_f, 50_	Stiffness Modulus of the Critical Point/MPa	N_tf_	Percent/%
SMA-13 SBS	4363.32	2181.66	560,180	1140.53	604,333	26
SMA-13 SBS BF	4917.05	2458.50	179,880	684.80	1,124,890	14
SUP-20 SBS	2980.44	1490.22	34,400	568.43	49,100	19
SUP-20 SBS BF	3648.8	1824.4	99,740	584.27	154,880	16

## Data Availability

The data presented in this study are available on request from the corresponding author.

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
