# Peer review of "Investigation on Fatigue Performance of Asphalt Mixture Reinforced by Basalt Fiber"

_materials, 2021, doi:10.3390/ma14195596_

Round 1
Reviewer 1 Report
- Try not to use acronyms in the abstract. In general, the abstract looks good in present form.
- Your introduction needs more information about asphalt mixture performance. I missing some general information regarding the performance of asphalt mixtures such as doi.org/10.1080/14680629.2017.1283353 and doi.org/10.1080/14680629.2021.1908408.
- Line 27: Avoid this kind of unprofessional language: "there are more and more vehicles". You can say "increased number of vehicles" or something like this.
- Please state clearly your objectives.
- Table 4: please include more data (ex. VMA, VFA, etc)
- Are there any disadvantages of using these fibers? Please include it in the conclusion section.
Reviewer 2 Report
Fatigue resistance is undoubtedly one of the most important performance characteristics of road pavement. The author in his study offers to improve fatigue resistance with basalt fiber, which is a very good and competitive solution. It would be good to note that the sustainability of this solution will be assessed in future studies using LCA and LCCA criteria. The properties of selected aggregates and asphalt concrete cannot be found in the article. In particular, the porosity of asphalt concrete has a significant effect on fatigue resistance, so it would be good to indicate this. In the introductory part, I recommend to mention HMAC (High modulus asphalt concrete) type as one of the world wide existing technological solutions for asphalt concrete design with high fatigue resistance, referring for example to DOI: 10.1080/10298436.2020.1850721, where fatigue resistance and the effect of RAP on fatigue and stiffness properties is studied.
Reviewer 3 Report
The article is interesting. The structure of the article is correct and meets the writing standards of scientific articles in international journals. Before publishing, the article should be modified in line with the comments below.
- You quote 8 articles of your own and many others from your country. Perhaps it is worth referring to other issues in the introduction:
- quiet surfaces
- Materials, (2020) 13 (3), number 675
- Construction and Building Materials (2020), vol. 263, number 120626
- Non-skid surfaces, heat islands and innovative surfaces
- Procedia Soc. Behav. Sci (2013), vol. 96 pp. 2745-2755
- Construction and Building Materials (2016), vol. 109, pp. 1-7
- Construction and Building Materials (2017), vol. 135, pp. 104-111
- Road Materials and Pavement Design, (2020) Vol. 21 Issue 8, pp. 2302-2320
- Line 107 - what does 70 # mean?
- In all the graphs in Fig. 3 it is necessary to explain what is marked with colors and what is marked with red triangles.
- Explain on what basis equation 1 was adopted.
- Figure 5d presents the samples A0, A1,…, C2? If so, mark them. If not, haw to explain the equations in table 5?
- Table 5 contains regression equations. Demonstrate statistical significance check for these equations.
- In Figure 9 remove all captions, change them into English. Demonstrate statistical significance check for these equations.
Round 2
Reviewer 1 Report
Thank you for addressing my comments.
Reviewer 3 Report
I thank the author for corrections and changes to the article that helped to improve it. I therefore consider that the article can be published.